# Using Flavonoid Substitution Status to Predict Anticancer Effects in Human Melanoma Cancers: An In Vitro Study

**DOI:** 10.3390/cancers16030487

**Published:** 2024-01-23

**Authors:** Katarzyna Jakimiuk, Łukasz Szoka, Arkadiusz Surażyński, Michał Tomczyk

**Affiliations:** 1Department of Pharmacognosy, Faculty of Pharmacy with the Division of Laboratory Medicine, Medical University of Białystok, ul. Mickiewicza 2a, 15-230 Białystok, Poland; michal.tomczyk@umb.edu.pl; 2Department of Medicinal Chemistry, Euroregional Center of Pharmacy, Faculty of Pharmacy with the Division of Laboratory Medicine, Medical University of Białystok, ul. Mickiewicza 2d, 15-222 Białystok, Poland; lukasz.szoka@umb.edu.pl (Ł.S.); arkadiusz.surazynski@umb.edu.pl (A.S.)

**Keywords:** flavonoids, melanoma, structure–activity relationship

## Abstract

**Simple Summary:**

The rising incidence of skin cancer, particularly melanoma, over the last few decades requires natural-product-based treatments with a lack of side effects. To evaluate the anticancer potential of 37 structurally diverse flavonoids, cytotoxic tests, DNA biosynthesis inhibition levels, and apoptosis pathways involved in cell death were investigated in A375 and C32 melanoma cells. Compounds **1**, **6**, **15**, and **37** reduced the viability of cell lines via the intrinsic (caspase-9) and extrinsic (caspase-8) pathways of apoptosis, where **16** and **17** mainly influenced DNA biosynthesis in cancer cells. In our study, we performed an anticancer analysis of compound **15** (5,6-dihydroxyflavone) for the first time. Although our data provide new insights into the molecular mechanisms of flavonoid-induced toxicity, in vivo studies are needed to assess their anti-cancer activities, especially flavonoid-core-containing hydroxyl groups.

**Abstract:**

Skin cancers are a dominant type of cancer that impacts millions per year. Cancer is a heterogeneous disease triggered by the irreversible impairment of cellular homeostasis and function. In this study, we investigated the activity of 37 structurally diverse flavonoids to find potentially active substances using two melanoma cell lines: C32 and A375. First, the cytotoxic potential and DNA biosynthesis inhibition of flavonoids were tested to determine the most active compounds in cancer and normal cells. Second, the molecular mechanism of the anticancer activity of flavonoids was elucidated using Western blot and immunofluorescence analyses. Compounds **1**, **6**, **15**, and **37** reduced the viability of A375 and C32 cell lines via the intrinsic and extrinsic pathways of apoptosis, whereas **16** and **17** acted in a higher degree via the inhibition of DNA biosynthesis. In our experiment, we demonstrated the anticancer activity of compound **15** (5,6-dihydroxyflavone) for the first time. The in vitro studies pointed out the importance of the flavonoid core in hydroxyl groups in the search for potential drugs for amelanotic melanoma.

## 1. Introduction

The most extensive organ of the human body is the skin, and skin cancers are a dominant type of cancer that impacts millions of people per year. Over the last few decades, the rising incidence of skin cancer has prompted the search for natural-product-based treatments, which might be the basis for producing synthetic and semi-synthetic drugs. This is particularly important in the case of melanoma, which is the most deadly skin cancer [1]. Malignant melanoma is one of the most aggressive malignant tumors and its extensive spread makes the mortality rate of patients with human malignant melanoma very high [2]. Although the surgical excision of the lesion followed by adjuvant treatment (chemotherapy and immunotherapy) are the two primary methods used in the treatment of malignant melanoma, both generate off-target outcomes that meaningfully reduce the quality of life of patients [3,4]. Therefore, experimental strategies concentrate on new therapies that avoid the undesirable effects of a drug (e.g., fotemustine, docetaxel, cisplatin) or medical treatment [5,6]. Nowadays, various preparations available on the market contain naturally derived compounds for use in the treatment of skin cancer or pre-cancerous conditions, such as Curaderm^®^, Cansema^®^, or birch bark ointment [7]. It has been recognized that both in vitro and in vivo studies have demonstrated that flavonoids can prevent carcinogenesis and might inhibit the growth of cancer cells. Flavonoids are low-molecular-weight secondary metabolites composed of two benzene rings and one heterocyclic ring and remain one of the most extensively studied polyphenolic compounds for their anticancer activity [8,9]. For example, flavonoids reduce the proliferation of cancer cells or mitochondrial membrane potential and increase autophagy, intracellular ROS (reactive oxygen species), and the expression of caspases, including caspase-3, -8, -9, and -10 [1]. The general activity of polyphenols is connected with the reactivity of their hydroxyl groups. The OH groups are able to dissociate to negatively charged phenolate ions and interact with proteins by forming hydrogen bridges or ionic bonds with positively charged molecules [10]. Flavonoids’ anticancer activity has been attributed to their ability to modulate oxidative stress through chelating with transition metals (Fe^2+^, Fe^3+^, Cu^2+^), which are involved in reactions generating free radicals [11,12]. Thus, in this study, we investigated the activity of 37 structurally diverse flavonoids (**1**–**37**) to find potentially active substances using two melanoma cell lines: primary human malignant amelanotic melanoma (A375 and C32) cells and normal skin fibroblasts. Extensive biological in vitro studies led to identifying which flavonoid substitution patterns play a crucial role in their anticancer activity, which allowed for the demonstration of the structure–activity relationships (SARs). Thus, in this study, we investigated the anti-cancer potential of 37 flavonoids to identify active compounds: apigenin (**1**), apigenin 8-*C*-glucoside (vitexin) (**2**), apigenin 7-*O*-glucoside (**3**), apigenin 7-*O*-glucuronide (**4**), apigenin 7-*O*-rutinoside (**5**), luteolin (**6**), luteolin 7-*O*-glucoside (cynaroside) (**7**), luteolin 7-*O*-glucuronide (**8**), luteolin 7-*O*-sambubioside (**9**), 5,7,3′-trihydroxy-4′-acetoxy-flavone-8-*C*-xyloside-2″-*O*-glucoside (scleranthoside A) (**10**), 5,7,3′-trihydroxy-4′-methoxyflavone-8-*C*-xyloside-2″-*O*-glucoside (scleranthoside B) (**11**), 5,7-dihydroxy-3′-methoxy-4′-acetoxyflavone-8-*C*-xyloside-2″-*O*-(4‴-acetoxy)-glucoside (scleranthoside C) (**12**), 5,7-dihydroxy-3′-methoxy-4′-acetoxyflavone-8-*C*-arabinoside-2″-*O*-(4‴-acetoxy)-glucoside (scleranthoside D) (**13**), chrysin (**14**), 5,6-dihydroxyflavone (**15**), 7,8-dihydroxyflavone (**16**), zapotin (**17**), kaempferol (**18**), 8-methoxykaempferol (**19**), kaempferol 3-*O*-glucoside (astragalin) (**20**), kaempferol 3-*O*-glucuronide (**21**), kaempferol 3-*O*-galactoside (hyperoside) (**22**), icariin (**23**), quercetin (**24**), 7-methoxyquercetin (rhamnetin) (**25**), 3′-methoxyquercetin (isorhamnetin) (**26**), quercetin 3-*O*-glucuronide (**27**), quercetin 3-*O*-rutinoside (rutin) (**28**), quercetin 3-*O*-rutinoside-7-*O*-glucoside (**29**), myricetin (**30**), tricin (**31**), robinetin (**32**), hesperetin (**33**), hesperetin 7-*O*-rutinoside (hesperidin) (**34**), hesperetin 7-*O*-neohesperidoside (neohesperidin) (**35**), daidzein (**36**), and genistein (**37**). The chemical and biological properties of flavonoids and their derivatives are linked with their structure and the position of different substitutions on the molecule. The position of the heterocyclic pyran ring in the structure, as well as the presence of hydroxyl groups and double bonds, allows for division into groups. Following this, compounds **1**–**17** belong to flavones, **18**–**32** belong to flavonols (additional OH group at C3), **33**–**35** belong to flavanols (lack of the double bond between C2 and C3), and **36**, **37** belong to isoflavones (heterocyclic pyran ring is attached at C3). In our study, first, cytotoxic potential and DNA biosynthesis inhibition were tested to determine the most active substances using A375 and C32 lines. Second, the molecular mechanism of the anticancer activity mediated by flavonoids was elucidated using Western blot and immunofluorescence analyses.

## 2. Materials and Methods

### 2.1. Chemicals and General Experimental Procedures

Compounds **1** (CAS number: 520-36-5), **2** (CAS number: 3681-93-4), **14** (CAS number: 480-40-0), **18** (CAS number: 520-18-3), **24** (CAS number: 6151-25-3), **25** (CAS number: 90-19-7), **26** (CAS number: 480-19-3), **30** (CAS number: 529-44-2), **33** (CAS number: 69097-99-0), and **37** (CAS number: 446-72-0) were purchased from Carl Roth (Karlsruhe, Germany); **15** (CAS number: 6665-66-3) was obtained from ThermoFisher Scientific (Waltham, MA, USA); **8** (CAS number: 29741-10-4), **16** (CAS number: 38183-03-8), 23 (CAS number: 489-32-7), **35** (CAS number: 13241-33-3), and **36** (CAS number: 552-66-9) were obtained from Cayman Chemical (Ann Arbor, MI, USA); and **4** was obtained from AmBeed (Arlington, IL, USA). Flavonoids **3** [13], **6** [14], **7**, **9** [15], **10**–**12** [16], **13** [17], **19** [18], **20**, **28** [19], **22**, and **29** [20] were isolated and identified in the Department of Pharmacognosy at the Medical University of Białystok (Poland). Compounds **5** (CAS number: 77298-67-0), **17** (CAS number: 14813-19-5), and **31** (CAS number: 520-32-1) were obtained from Biosynth Carbosynth (London, UK); **21** (CAS number: 22688-78-4) and **27** (CAS number: 22688-79-5) were purchased from Extrasynthese (Genay, France); and **32** (CAS number: 490-31-3) and **34** (CAS number: 520-26-3) were obtained from Fluka Analysis (St. Louis, MO, USA). The cell viability and proliferation assays were performed on the melanoma cell lines C32 (CRL-1585) and A375 (CRL-1619) and human skin fibroblasts (CCD25Sk). The cells were obtained from ATCC (American Type Culture Collection, Manassas, VA, USA) and cultured in DMEM (Dulbecco’s Modified Eagle’s Medium, ThermoFisher Scientific, 41966029) supplemented with 10% FBS (fetal bovine serum, ThermoFisher Scientific, 10270106), 100 U/mL penicillin, and 100 U/mL streptomycin (ThermoFisher Scientific, 15140122). The cells were incubated at 37 °C in the air with 5% CO_2_. PBS (phosphate-buffered saline, 14190136), the Dead Cell Apoptosis Kit for flow cytometry (#V13242), a protease and phosphatase inhibitor cocktail (#78440), and neutral red (N3246) were purchased from ThermoFisher Scientific. MTT (3-(4,5-dimethyl-2-thiazolyl)-2,5-diphenyl-2H-tetrazolium bromide) (M5655), SDS (sodium dodecyl sulfate, L3771), NaOH (sodium hydroxide, 655104), anti-actin antibody (#A2066, 1:2000), skim milk (anti-mouse IgG-HRP, #A9044, 1:5000 or anti-rabbit IgG-HRP, #A9169, 1:5000), and cisplatin (cPt), which was used as a positive control (P4394), were purchased from Sigma-Aldrich (Saint Louis, MO, USA). The [^3^H]-thymidine (#MT6037) was obtained from Hartmann Analytic for the proliferation assay. The Mini-Protean Tetra Cell system, nitrocellulose membranes, and Mini Trans-Blot Cell wet blotting system were purchased from Bio-Rad (Hercules, CA, USA). The Cell Lysis Buffer (#9803), anti-caspase-9 antibody (#9508), anti-caspase-8 antibody (#9746), anti-caspase-3 antibody (#9662), anti-caspase-7 antibody (#12827), anti-PARP antibody (anti-poly-(ADP-ribose) polymerases antibody) (#9542), anti-RSK2 antibody (rabbit polyclonal antibody) (#5528), and anti-pSer386-RSK2 antibody (#11989) were obtained from Cell Signaling Technology (Danvers, MA, USA). The ECL-HRP substrate (enhanced chemiluminescence-horseradish peroxidase substrate) was obtained from GE Healthcare (Chicago, IL, USA).

### 2.2. Cell Viability Assays

#### 2.2.1. 3-(4,5-Dimethyl-2-thiazolyl)-2,5-diphenyl-2H-tetrazolium Bromide (MTT) Assay

The MTT colorimetric assay was performed as a preliminary evaluation of the cytotoxicity of the selected flavonoids according to the method described and validated by Strawa et al. [21]. Briefly, melanoma cells and fibroblasts were seeded in 96-well microplates at a density of 1 × 10^4^ cells per well. Fibroblasts were used as a control cell line in order to assess the selectivity of the compounds. Following 24 h of incubation (at 37 °C with 5% CO_2_), the cells were exposed to different concentrations of compounds **1**–**37** and cisplatin (6.25, 12.5, 25, 50, 100, or 200 μM) for 24, 48, and 72 h. Subsequently, the cells were washed with PBS, incubated with MTT solution for 2 h, and lysed with dimethyl sulfoxide (DMSO) supplemented with 1% (*v*/*v*) Sorensen’s glycine buffer. The absorbance was measured using a microplate reader (EPOCH 2, BioTek, Winooski, VT, USA) at a wavelength of 570 nm. The compounds were used at a final DMSO concentration of no more than 0.1% (*v*/*v*) in each well. The results were generated from at least three independent experiments. Cisplatin (cPt) was used as a positive control. The IC_50_ values were calculated with GraphPad Prism 9 software (GraphPad Software, San Diego, CA, USA).

#### 2.2.2. Neutral Red Uptake (NRU) Assay

The NRU (neutral red uptake) was conducted as previously described by Szoka and co-workers, with some modifications [22]. Briefly, melanoma cells and fibroblasts were seeded in microplates at the density of 1 × 10^4^ cells per well and allowed to adhere for 24 h. Then, cells were incubated with different concentrations of compounds **1**–**37** and cPt (6.25, 12.5, 25, 50, 100, or 200 μM) for 24, 48, and 72 h. After treatment, the culture medium was replaced with a neutral red solution (40 µg/mL). An additional 2 h of incubation was performed. The neutral red solution was then removed and the neutral red accumulated in the cells was dissolved in 50% EtOH with 1% of the glacial acetic acid. The absorbance was estimated at 540 nm and IC_50_ values were calculated using GraphPad Prism software (GraphPad Software, San Diego, CA, USA).

### 2.3. Proliferation Assay

The inhibition of DNA biosynthesis was evaluated using [^3^H]-thymidine incorporation into the DNA of melanoma cells (C32, A375) and fibroblasts treated with the studied compounds [21]. The cells were briefly seeded in 24-well plates (1 × 10^5^ cells per well) and allowed to adhere for 24 h. Subsequently, the studied cells were treated with 1 mL of DMEM with 0.5 µCi [^3^H]-thymidine (6.7 Ci/mmol) and various concentrations (6.25–200 µM) of the most active compounds. Cisplatin (cPt) was used as a positive control. After incubation for 24, 48, and 72 h, the cells were rinsed three times with PBS. Then, cells were lysed in 1 mL of 0.1 M NaOH containing 1% SDS. The cell lysate was moved into scintillation vials containing scintillation liquid, and the radioactivity of the samples was measured using a liquid scintillation analyzer (Perkin-Elmer, Waltham, MA, USA).

### 2.4. Apoptosis Assay

The apoptosis assay was conducted as described before [22]. Firstly, cells were placed in 6-well plates at a density of 1 × 10^5^ cells per well and allowed to adhere for 24 h. Then, cells were treated with 25 µM, 50 µM, and 100 μM of the selected flavonoids (**1**, **6**, **15**, **16**, **17**, **37**) and incubated for 48 h. Floating and adherent cells were collected and assayed using the manufacturer’s protocol supported by the Dead Cell Apoptosis Kit for flow cytometry. Concisely, 100 μL of an annexin-binding buffer containing a 5 μL annexin V-FITC conjugate solution and 1 μg/mL PI (propidium iodide) was used to disperse the cells. After 15 min of incubation, an additional 400 μL of the mentioned buffer was added. The cells were analyzed with a DxFLEX flow cytometer (Beckman Coulter, Brea, CA, USA).

### 2.5. Western Immunoblot

Floating cells were harvested by centrifugation of the conditioned media (300× *g*, 5 min). Cell pellet and adherent cells were lysed in a cell lysis buffer supplemented with inhibitors of proteases and phosphatases. The cell lysate was sonicated (2 × 10 s) on ice with an ultrasonic homogenizer (UP200S, Hielscher Ultrasonics GmbH, Teltow, Germany) and centrifuged for 10 min at 10,000× *g*. The supernatant was collected and stored in a freezer at −70 °C. Protein concentrations in the cell supernatants were determined using a Lowry assay. Obtained proteins were dissolved on SDS-PAGE gels (7.5%, 10%, or 12%) using the Mini-Protean Tetra system. The proteins were transferred to nitrocellulose membranes using the Mini Trans-Blot Cell wet blotting system. The 5% skim milk was used to block membranes. Then, membranes were probed for 24 h at 4 °C with the following antibodies: anti-phospho-RSK, anti-RSK, anti-PARP, anti-cleaved PARP, anti-caspase-8, anti-caspase-9, anti-caspase-7, anti-caspase-3, and anti-actin. Membranes were incubated with an ECL-HRP (enhanced chemiluminescence-horseradish peroxidase) substrate, and the signal was spotted using the BioSpectrum Imaging System and VisionWorks LS software 7.0 (Ultra-Violet Products, Ltd., San Gabriel, CA, USA). The intensity of the bands was quantified by densitometric analysis using UVIBand software, version 15.08d (Uvitec Ltd., Cambridge, UK).

### 2.6. Immunofluorescence Microscopy

Firstly, to examine fluorescent signals, cells were placed in the bottom of 96-well black plates at 1 × 10^4^ cells per well and permitted to adhere for 24 h. Subsequently, the cells were treated with compounds **1**, **6**, **15**–**17**, and **37** at concentrations of 25 μM, 50 μM, or 25 μM cPt for 48 h. After incubating and removing the medium, cells were fixed in 4% paraformaldehyde for 15 min. Later, cells were rinsed three times with PBS and permeabilized with 0.1% Triton X-100 for 5 min at room temperature. The 10% heat-inactivated goat serum in the PBS was used to block non-specific binding. Subsequently, cells were treated with the following antibodies overnight: anti-cleaved caspase 3 and anti-cleaved caspase-7. Next, cells were incubated with the Alexa Fluor 594 conjugated antibody for 1 h in a dark place. Cell nuclei were stained with Hoechst 33342 and F-actin was stained with Phalloidin-Atto 488. Fluorescent signals were captured using a BD Pathway 855 confocal microscope (Becton Dickinson, Franklin Lakes, NJ, USA). The mean fluorescence intensity was calculated using ImageJ software (National Institutes of Health, Bethesda, Rockville, MD, USA).

### 2.7. Statistical Analysis

The statistical analysis for the IC_50_ calculations was performed using nonlinear regression using GraphPad Prism 9 software (GraphPad Software, San Diego, CA, USA) and MS Excel 2019 software (Microsoft, Washington, DC, USA). All data were received as mean ± SD (from at least three independent replicates), and the significance of differences was determined using ANOVA.

## 3. Results

### 3.1. Cell Viability Assays

In the first step, the potential cytotoxic effect of different concentrations of the selected flavonoids (Table 1, Figure 1) and cPt, a well-known anticancer agent, was studied using MTT and NRU tests on A375 and C32 melanoma cell lines and fibroblasts (Table 2 and Table 3, Appendix A).

An MTT assay was carried out separately for each sample for a preliminary evaluation of the cytotoxic effects of all samples on the viability of A375, C32, and fibroblast cells after 24, 48, and 72 h of incubation. The cytotoxic potential was expressed as a median inhibitory concentration (IC_50_) value. At this point, six compounds showed the highest activity in both A375 and C32 cell lines: **1**, **6**, **15**, **16**, **17**, and **37**. Compound **17** (zapotin) displayed the strongest and a selective effect among all tested compounds after 48 h of incubation, where the IC_50_ reached values of 31.1 ± 2.9, 26.5 ± 1.3, and 78.0 ± 5.2 µM for A375, C32, and normal cells, respectively. The cytotoxic activity of the most active compounds increased when the incubation time was increased from 24 h to 48 h. Nevertheless, this effect was not consistent when treatment was increased further from 48 h to 72 h. On the other hand, sixteen compounds (**2**, **3**, **8**–**13**, **21**–**23**, **27**, **29**, **34**–**36**) did not indicate any activity, even at the highest used concentration (200 µM).

Despite the fact that the MTT test is the most common method for the evaluation of cell viability, several studies have revealed inaccuracies that are inherent in this method. Some experiments demonstrate that flavonoids are redox-active against tetrazolium salt even in the absence of cells, which interferes with the MTT test [15]. Thus, we performed an NRU assay to confirm the reliability of the obtained results with the MTT method. The data obtained from the NRU analysis indicate a strong activity of the six compounds mentioned in MTT test. After 48 h of exposure, in both A375 and C32 cells, the IC_50_ of **17** (14.7 ± 1.2 and 14.7 ± 1.1 µM, respectively) showed comparable cytotoxicity to cPt (14.4 ± 1.1 and 15.0 ± 1.2 µM). Importantly, 5,6-dihydroxyflavone (**15**), 7,8-dihydroxyflavone (**16**), and zapotin (**17**) were more active in cancer cells than in fibroblasts.

### 3.2. Proliferation Assay

The observed decrease in cancer cell viability could have been the result of an impaired cell proliferation rate. Therefore, based on the cytotoxic tests, we selected the most active flavonoids (**1**, **6**, **15**–**17**, **37**) and performed a proliferation assay (Table 4). Treatment with **17** showed a significant decrease in [^3^H]-thymidine incorporation into newly synthesized DNA after 48 h of incubation in both A375 and C32 cell lines (IC_50_ = 6.0 ± 0.2 and 9.1 ± 1.2 µM, respectively). It is worth mentioning that **17** affected DNA biosynthesis more in cancer cells than in normal cell lines.

### 3.3. Apoptosis Assay

To further explore the mechanism by which these compounds affect cell viability, dual staining with annexin V and propidium iodide followed by flow cytometry was performed after 48 h of treatment of melanoma cells with compounds **1**, **6**, **15**–**17**, and **37** at concentrations of 25, 50, and 100 µM (Figure 2). The data showed that these flavonoids induced apoptosis in melanoma cells. Among them, **6**, **15**, **16**, and **17**, even at the lowest concentration (25 µM), highly increased the levels of early and late apoptotic cells to 63.6 ± 6.1% and 13.0 ± 1.8% (**6**), 42.0 ± 4.5% and 8.9 ± 2.2% (**15**), 49.1 ± 3.9% and 6.4 ± 1.5% (**16**), and 48.0 ± 3.9% and 13.2 ± 1.9% (**17**), respectively. Further, the 50 µM level of early and late apoptotic cells was elevated after using **1** and **37** to 34.2 ± 2.8% and 7.1 ± 1.6% (**1**) and 38.1 ± 4.2% and 8.9 ± 1.9% (**37**). Thus, the percentage of apoptotic A375 cells increased as follows: **1**–**37** < **15**–**16**–**17** < **6**. On the other hand, in the C32 melanoma cell line, the most significant increase in the percentage of early and late apoptotic cells after treatment with a 25 µM concentration was observed for **17** and **1**: 19.7 ± 1.6% and 32.8 ± 2.4% (**17**) and 13.3 ± 2.4% and 29.4 ± 3.3% (**1**). Therefore, the rise in the percentage of apoptotic C32 cells increased as follows: **37**–**16** < **6**–**15** < **1**–**17**.

In the next step, the levels of the essential mediators of apoptosis were analyzed by the Western blot method (Figure 3, Appendix A). The treatment of A375 cells with **1**, **6**, **37**, and cPt for 48 h resulted in the cleavage of both initiator caspases, caspase-9 and caspase-8, and decreased levels of the caspases-3 and -7 zymogens. However, compound **6**’s action was inversely proportional to the used concentration. Treatment with compounds **15** and **16** increased the expression of cleaved caspase-8, but **16** induced a weak cleavage of caspase-9. On the other hand, after using **17**, there were no visible signs of active initiator caspases. However, the expression of caspase-8 pro-form was significantly downregulated. Compounds **16** and **37**, at higher concentrations, and cPt impaired the phosphorylation of the RSK protein (ribosomal kinase).

In the C32 cell line, flavonoids **1**, **6**, **15**, and **37** and cPt generated cleaved PARP and caspase-9. The expression of caspase-8 zymogen was below the limit of detection in these cells. Furthermore, **17**, **37**, and cPt inhibited the phosphorylation of RSK.

In Figure 4 and Appendix A, the expression of cleaved caspase-3 and cleaved caspase-7 revealed by immunofluorescent staining in both melanoma cell lines is presented. In the A375 cell line, 48 h of treatment with **1** and **6** slightly increased the level of cleaved caspase-3. Furthermore, **1**, **6**, **15**, and **37** notably extended caspase-3 expression in the C32 cells. Also, the level of the cleaved caspase-7 slightly increased in C32 cells after treatment with **6**, **16**, and **17**.

## 4. Discussion

Despite the fact that surgery, radiation therapy, and chemotherapy are conventionally used to treat cancers, there is undoubtedly a great need to investigate natural products that may reduce the frequency of cancer incidences or the side effects of synthetic drugs [23].

The results available in the literature that describe the anticancer activity of flavonoids are diverse and sometimes contradictory. This dissimilarity may be connected with various experimental conditions (e.g., concentrations of substrates or conditions of incubation) or the limited number of compounds included in such studies. Thus, in this study, we tested a large set of flavonoid compounds for their cytotoxic, antiproliferative, and apoptotic effects under the same conditions to obtain values that are as close to those in the real world as possible. Compound **1** (apigenin), which may be considered the major flavone in the plant kingdom, is also one of the most studied phytoconstituents for its anticancer activity. Zhao and co-workers investigated the cytotoxic effect of apigenin using MTT assays. According to their results, apigenin at a concentration of 40 µM decreased A375 cell viability [24], which is in agreement with our results after 48 h of incubation (IC_50_ = 31.9 ± 1.0 µM). On the other hand, Cao et al. implied that apigenin at 10 and 20 μM significantly inhibited A375 cell motility and invasion in a dose-dependent manner [25]. Even though MTT seems to be the most common test for detecting cytotoxicity, no data are available for the C32 cell line using this method thus far. Nevertheless, the SRB test showed that **1** is a more potent cytotoxic compound against C32 cells than against A375 cells [26]. These results follow our NRU assays. To determine structure–activity relationships as predictors of cytotoxicity, we compared the cytotoxic effects of closely related compounds. Another widely distributed flavone aglycone in plants is luteolin (**6**), which possesses one more OH group in the B-ring. According to our data, it affected A375 and C32 cell viability in both MTT (IC_50_ = 38.23 and 42.10 µM, respectively, after 48 h of incubation) and NRU (IC_50_ = 25.70 and 13.11 µM, respectively, after 48 h of incubation) tests; hence, **6** is a powerful cancer-inhibiting compound. In previous studies, luteolin possessed IC_50_ values that reached 18.6 µg/mL (IC_50_ = 65.0 μM, A375, 48 h of incubation) and 95.1 µg/mL (IC_50_ = 332.5 μM, C32, 24 h of incubation) [1]. In our study, luteolin (**6**) treatment for 48 h decreased A375 cell viability, with IC_50_ values of 38.2 μM and 25.7 μM, as determined by MTT and NRU assays, respectively. The IC_50_ values for **6** in C32 cells after 24 h treatment were very diverse depending on the assay used, reaching >200 μM in the MTT test and 41.3 μM in the NRU assay. Nevertheless, comparing compounds **15**, **16**, and **17** seems particularly promising for further stages of research. In our experiment, we demonstrated the anticancer activity of **15** (5,6-dihydroxyflavone) for the first time. This is extremely important because the hydroxyl groups in the A-ring of flavonoids may be responsible for their activity as well as for sharing the unsubstituted B ring, which was also confirmed by the high activity of **16** (7,8-dihydroxyflavone) (Table 2 and Table 3). In contrast to **15**, flavonoid **16** has already been reported as an anticancer agent in two human malignant melanoma cell lines (SK-MEL-2 and G-361). In the MTS test, the inhibition of the viability by 7,8-dihydroxyflavone (**16**) was significant in both melanoma cell lines (IC_50_ values were in the range of 200–250 µM) [27]. Moreover, **17**, selected in our survey as the most active in the cell viability test, is mentioned in only a few studies for its anticancer activity (e.g., HeLa, SCC, MCF-7, U251N, and PANC-1 cell lines) [21,28,29,30]. It is worth mentioning that increasing activity against cancer cells with **15**–**17** is not connected to higher toxic effects against normal cells. Compound **37** differs in its structural structure from other active flavonoids; it belongs to the group of isoflavones that have the B-ring attached to the C-ring in position 3 [31]. Genistein (**37**) was a formerly mentioned inhibitor of the growth of another melanoma cell line (OCM-1) and in many other cancers, such as leukemia, lymphoma, ovarian, cervical, leiomyoma, melanoma, neuroblastoma, gastric, pancreatic, breast, and prostate cancer cells [32,33,34].

Reviewing the results obtained from the MTT and NRU tests, we concluded that flavonoid acetylation or glycosylation in carbons in the B-ring (e.g., **10**–**13**) and benzo-*γ*-pyrane ring (e.g., **2**–**5**, **27**–**29**) reduce their activity in melanoma cells. Our conclusions are consistent with the available literature data regarding flavonoids and their derivative biological activities. In silico molecular docking analysis and ligand–active site interactions were studied by Nile et al. using inter alia quercetin, quercetin glycoside, quercetin 3,4′-*O*-diglucoside and α-glucosidase, tyrosinase, and xanthine oxidase enzymes. Their results show that docked quercetin is active against glucosidase, tyrosinase, and xanthine oxidase, which is consistent with our in vitro study [35].

Furthermore, the methylation of hydroxyls is a typical modification found in flavonoid metabolites. Based on previous studies, higher methoxylated compounds are significantly more toxic than less methoxylated molecules. It was found that 7-methoxy-baicalein is more cytotoxic than baicalein. It is also assumed that the overall number of OH groups negatively correlates with the cytotoxic potential of the molecule (apigenin vs. quercetin) [36]. Nevertheless, it seems that the overall number of OH groups in the C- and B-rings unfavorably influences the cytotoxic potential of the compounds. There is a lack of information about the A-ring. According to our results, the appearance of free hydroxyls in the A ring (**15**, **16**) or the methylation of OH groups in the B-ring (**17**) increased their function as potential anticancer agents.

In the next step of our experiment, we evaluated the inhibition of DNA biosynthesis by measuring [^3^H]-thymidine incorporation into the DNA of C32, A375, and normal cells. Thymidylate deficiency and fluctuations in cellular nucleotide pools slow tumor growth by impairing DNA replication and repair [37,38]. Again, using **17** caused the most notable decrease in the assay after 48 h of incubation for both A375 and C32 cell lines (IC_50_ = 6.0 ± 0.2 and 5.3 ± 0.4 µM, respectively), with very low toxic effects against normal cells. Recently, Strawa and co-workers noticed that **17** evokes DNA biosynthesis inhibition in an oral squamous carcinoma (SCC-25) cell line [21].

In cancers, like many other diseases, an imbalance in apoptotic mechanisms occurs. Flavonoids are known as substances that can be aspirants for the ancillary therapy of cancer via modulating apoptotic pathways [39,40]. A summarized scheme of action by **1**, **6**, **16**–**17**, and **37** in A375 and C32 cells is given in Figure 5. A graphical representation of the substitutions and chemical groups relevant to the cytotoxic activity of these flavonoids is shown in Figure 6. It was previously reported that treatment with apigenin (**1**) induces the intrinsic and extrinsic pathways of apoptosis. As Bo et al. presented, increased levels of active caspase-3 and caspase-9 and the cleavage product of poly-(ADP-ribose) polymerase (cPARP) are present in human colon carcinoma HCT-116 cells treated with apigenin [41]. However, **1** activates kinase RSK, which is responsible for the inactivation of cytosolic pro-apoptotic proteins and the activation of transcription factors that mediate their synthesis [42,43]. This phenomenon may become one of the mechanisms by which melanoma cells defend themselves to evade apoptosis. In our study, **1** provoked the activation of caspases-8, -7, -9, and -3 in the A375 cells and caspase-3 in the C32 cells. Compound **6**, which differs from apigenin by the presence of a hydroxyl group at C4′, caused a dose-dependent activation of caspases in the A375 cell line. The lower concentration (25 µM) induced extrinsic (cleavage of caspase-8) and intrinsic (cleavage of caspase-3, -7, and -9) pathways of apoptosis. However, the presence of cleaved initiator caspases was not found in cells treated with a higher concentration of **6**. Also, in C32 cells, treatment with **6** did not cause an increase in the cleavage of initiator caspases. In contrast, it induced the processing of caspase-3 and caspase-7 and promoted apoptosis but inhibited RSK proteins. These results may suggest that **6** shows two dose-dependent mechanisms of action subsequent to cell death at the appointed time of incubation (48 h). Another study demonstrated the activation of caspases-3 and -9 in non-small-cell lung cancer cells A549 and H460 or oral squamous cancer cells SCC-4 [44,45,46]. Flavonoid **15**, a non-studied till now compound, induced caspase-8 activation in A375 melanoma cells. Furthermore, it possessed a bi-directional and dose-dependent action on RSK proteins, suppressing their phosphorylation at higher concentrations. On the other hand, **15** activated the intrinsic apoptosis pathway by the induction of caspase-9 in C32 cells. A previous investigation indicated that **16** is an activator of caspase-3 and c-PARP in SK-MEL-2 and G-361 melanoma cells [27]. Consistent with these findings, **16** induced apoptosis in both A375 and C32 cell lines. This has been formerly shown by Western blot analysis, in which **17** degraded PARP-1, a substrate of caspase 3, in HeLaPKCεA/E cells [47]. According to our immunofluorescence and Western blot results, treatment with **17** increased cleaved caspases-3 and -7 levels and led to the processing of PARP. The flow cytometric analysis of C32 melanoma cells showed that **17** (at 25 µM) caused the most significant increase in the percentage of early apoptotic cells (19.7 ± 1.6%) and the percentage of late apoptotic cells (32.8 ± 2.4%). However, **17** induced a massive RSK phosphorylation in A375 cells while inhibiting it in C32 cells. To summarize, the anticancer potential of **17** is most probably based on DNA biosynthesis inhibition (in both A375 and C32) and the inhibition of RSK activity (C32). As previous studies reported, genistein (**37**) acts on cancer progression mainly by targeting the NF-κB, Akt, and caspase pathways [48,49,50]. Based on our investigation, **37** led to apoptosis by the activation of caspases-3, -7, -8, and -9. Other studies of this compound reported that it could induce apoptosis in various human cancer cells (e.g., HT-29) via caspase-3 activation [34,51,52]. This may prove the importance of the B-ring attached to the C-ring in position 3 for cytotoxic activity. Indeed, in our study, the anticancer effects and their mechanism for compounds **37** and **1**, which differ in the positioning of the 4-hydroxyphenyl group, were similar. However, **37** and **1** inversely affected the phosphorylation of RSK.

In sum, the hydroxylation of the B-ring in the 3′ position does not seem to affect the strength and mechanism of the anticancer action of flavonoids (**1**; 5,7,3′-hydroxyflavone vs. **6**; 5,7,3′,4′-tetrahydroxyflavone). Our conclusions agree with those of previous studies [53]. Notably, many reports have provided evidence of the influence of flavonoid glycosylation on their activity [46]. Glycosylated flavonoids (**6** vs. **10**–**13** or **1** vs. **2**, **3**) possess weaker cytotoxic activity than aglycones. The weak antiproliferative effects of flavonoid glycosides might be derived from steric blocking involved in cell entry and receptor binding [47]. Flavonoid A-ring poly-*O*-methylation contributed to enhancing their biological activity [46], which is also consistent with our scores (e.g., **25** vs. **24**). It is worth noting that 5-OH seems to play a crucial role in membrane-dependent processes associated with the bioactivity of flavonoids [48]. In the present experiment, the most potent compounds have a hydroxyl group in the A rings at the C5 position (**1**, **6**, **15**, and **37**). However, further explanations should be provided.

## 5. Conclusions

The data presented above highlight the structure—activity relationship determination of the chemical substitutions of flavonoid compounds responsible for anticancer effects. The in vitro studies highlight the importance of the flavonoid core in hydroxyl groups. In conclusion, our results indicate that apigenin, luteolin, 5,6-dihydroxyflavone, and genistein reduce the viability of amelanotic melanoma A375 and C32 cell lines via the intrinsic and extrinsic pathways of apoptosis. On the other hand, 7,8-dihydroxyflavone and zapotin likely act after the inhibition of DNA biosynthesis. Our studies also indicate that the RSK pathway is involved in the response of melanoma cells to cytotoxic concentrations of certain flavonoids, which may provide a basis for studies on the combined use of flavonoids with inhibitors of RSK kinases. Nevertheless, additional investigations using in vivo assays to test these anticancer effects are required to fully understand their mechanism of action.

## Figures and Tables

**Figure 1 cancers-16-00487-f001:**
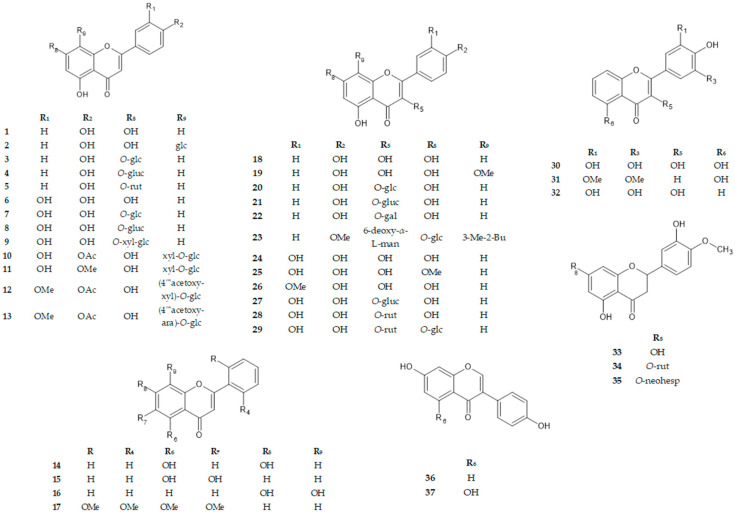
Molecular structures of the tested compounds. Me—methyl; Bu—butenyl; Ac—acetyl; glc—glucose; gluc—glucuronide; ara—arabinoside; man—mannose; rut—rutinose; xyl—xylose; neohesp—neohesperidoside.

**Figure 2 cancers-16-00487-f002:**
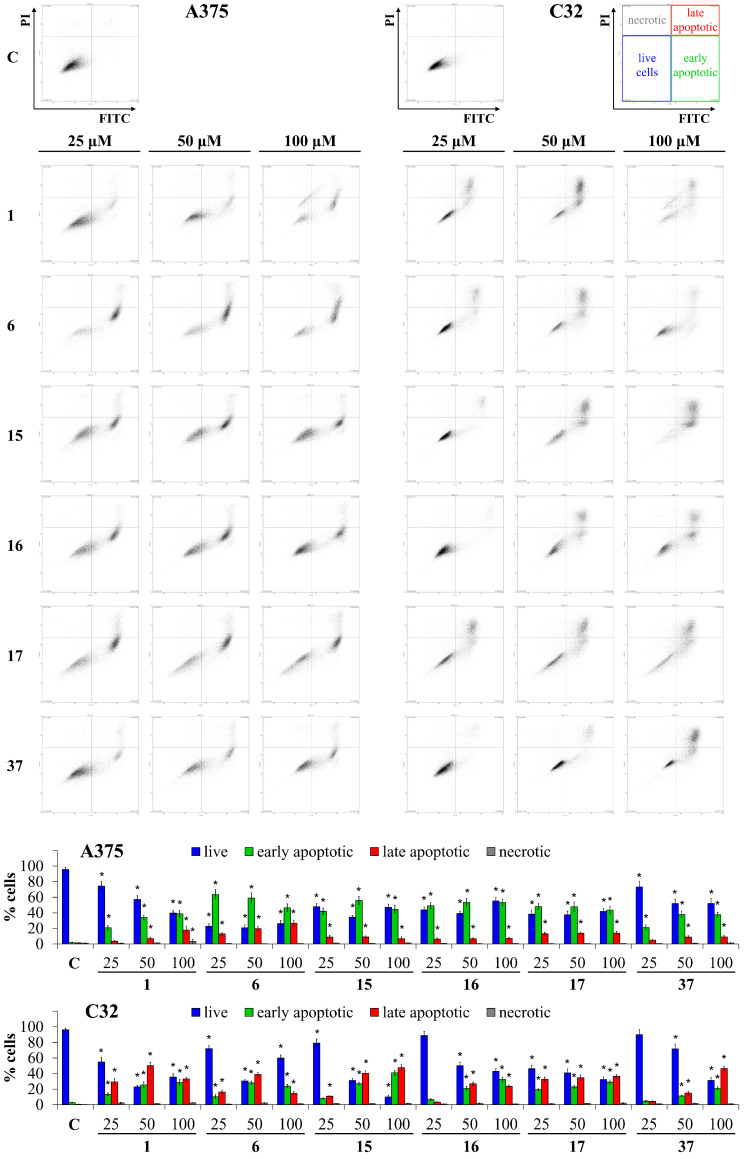
Flow cytometric analysis of A375 and C32 melanoma cells after incubation with compounds **1**, **6**, **15**–**17**, and **37** (25, 50, 100 µM) for 48 h and subsequent staining with annexin V-fluorescein isothiocyanate (*x*-axis) and PI (*y*-axis), C-control (cells treated with 0.1% DMSO). * *p* < 0.05, when compared to the control group.

**Figure 3 cancers-16-00487-f003:**
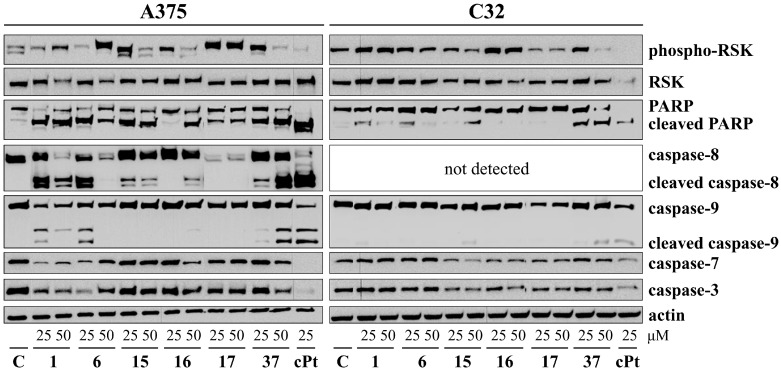
Western blot analysis of phospho-RSK, RSK, PARP, caspase-8, caspase-9, caspase-7, and caspase-3 in A375 and C32 cells treated with **1**, **6**, **15**–**17**, **37**, and cPt for 48 h. Actin served as a control for protein loading. C—control cells treated with 0.1% DMSO. The uncropped blots are shown in Appendix A.

**Figure 4 cancers-16-00487-f004:**
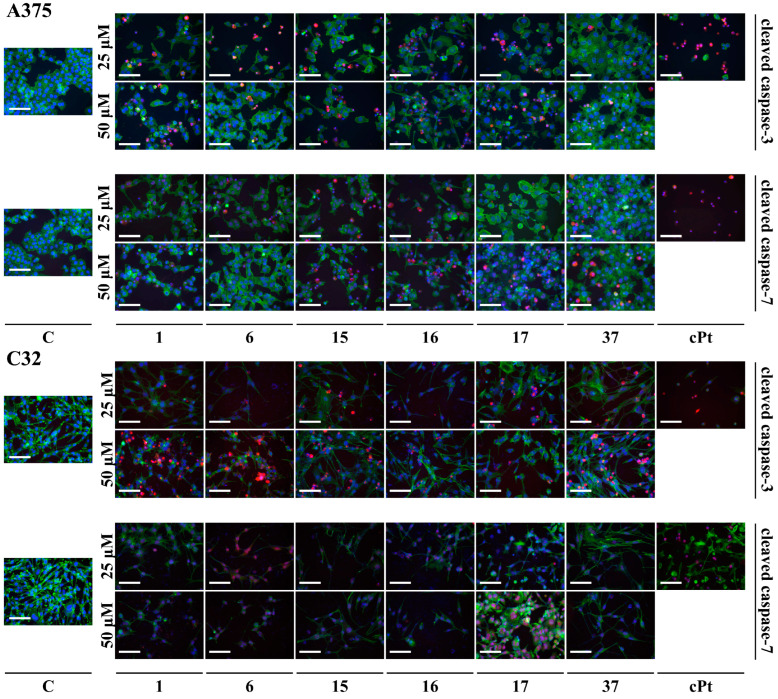
Immunofluorescence imaging of cleaved caspase-3 and cleaved caspase-7 (red fluorescence) in A375 and C32 cells treated with **1**, **6**, **15**–**17**, **37**, and cPt for 48 h. Cell nuclei were stained with Hoechst 33342 (blue fluorescence) and F-actin was stained with Phalloidin-Atto 488 (green fluorescence). Objective ×20. Scale bar 100 μm. C—control (cells treated with 0.1% DMSO).

**Figure 5 cancers-16-00487-f005:**
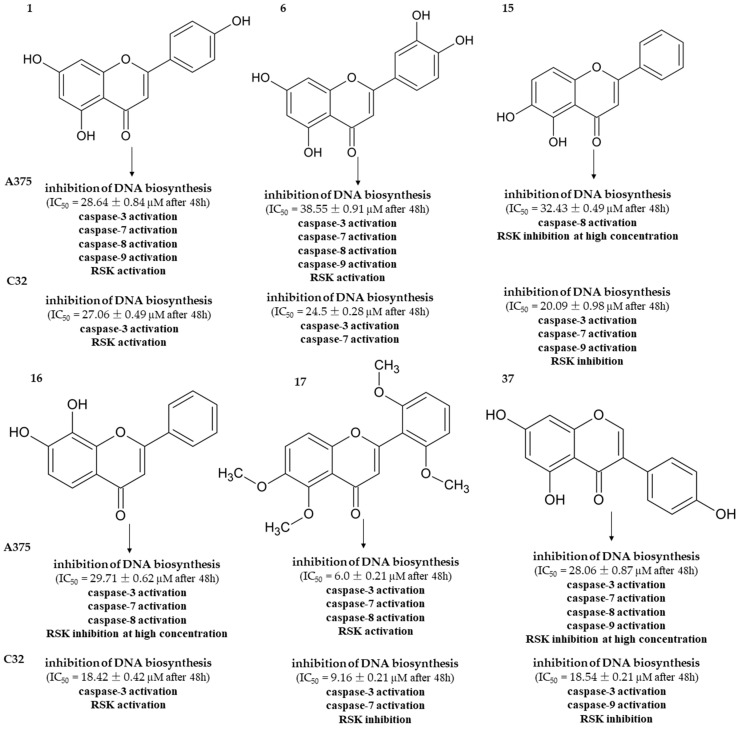
Proposed scheme of action by **1**, **6**, **16**–**17**, and **37** in A375 and C32 melanoma cells.

**Figure 6 cancers-16-00487-f006:**
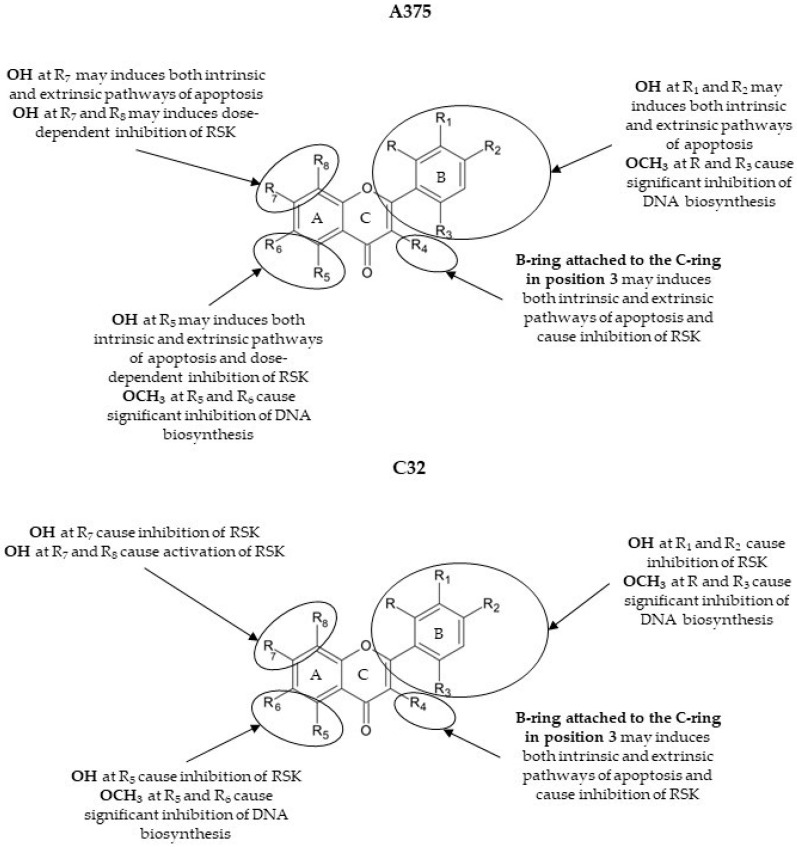
Chemical groups responsible for flavonoid activity (SAR, structure–activity relationship).

**Table 1 cancers-16-00487-t001:** Substitution pattern of compounds **1**–**37**.

No.	π C2-C3	B-Ring					Benzo-*γ*-Pyrane Ring				
		C2′ (R) r (R)	C3′ (R_1_)	C4′ (R_2_)	C5′ (R_3_)	C6′ (R_4_)	C3 (R_5_)	C5 (R_6_)	C6 (R_7_)	C7 (R_8_)	C8 (R_9_)
**1**	+	H	H	OH	H	H	H	OH	H	OH	H
**2**	+	H	H	OH	H	H	H	OH	H	OH	glc
**3**	+	H	H	OH	H	H	H	OH	H	*O*-glc	H
**4**	+	H	H	OH	H	H	H	OH	H	*O*-gluc	H
**5**	+	H	H	OH	H	H	H	OH	H	*O*-rut	H
**6**	+	H	OH	OH	H	H	H	OH	H	OH	H
**7**	+	H	OH	OH	H	H	H	OH	H	*O*-glc	H
**8**	+	H	OH	OH	H	H	H	OH	H	*O*-gluc	H
**9**	+	H	OH	OH	H	H	H	OH	H	*O*-xyl-glc	H
**10**	+	h	OH	OAc	H	H	H	OH	H	OH	xyl-*O*-glc
**11**	+	h	OH	OMe	H	H	H	OH	H	OH	xyl-*O*-glc
**12**	+	H	OMe	OAc	H	H	H	OH	H	OH	(4‴acetoxy-xyl)-*O*-glc
**13**	+	H	OMe	OAc	H	H	H	OH	H	OH	(4‴acetoxy-ara)-*O*-glc
**14**	+	H	H	H	H	H	H	OH	H	OH	H
**15**	+	H	H	H	H	H	H	OH	OH	H	H
**16**	+	H	H	H	H	H	H	H	H	OH	OH
**17**	+	OMe	H	H	H	OMe	H	OMe	OMe	H	H
**18**	+	H	H	OH	H	H	OH	OH	H	OH	H
**19**	+	H	H	OH	H	H	OH	OH	h	OH	OMe
**20**	+	H	H	OH	H	H	*O*-glc	OH	H	OH	H
**21**	+	H	H	OH	H	H	*O*-gluc	OH	H	OH	H
**22**	+	H	H	OH	H	H	*O*-gal	OH	H	OH	H
**23**	+	H	H	OMe	H	H	6-deoxy-*α*-L-man	OH	H	*O*-glc	3-Me-2-Bu
**24**	+	H	OH	OH	H	H	OH	OH	H	OH	H
**25**	+	H	OH	OH	H	H	OH	oH	H	OMe	H
**26**	+	H	OMe	OH	H	H	OH	OH	H	OH	H
**27**	+	H	OH	OH	H	H	*O*-gluc	OH	H	OH	H
**28**	+	H	OH	OH	H	H	*O*-rut	OH	H	OH	H
**29**	+	H	OH	OH	H	H	*O*-rut	OH	H	*O*-glc	H
**30**	+	H	OH	OH	OH	H	OH	OH	H	OH	H
**31**	+	H	OMe	OH	OMe	H	H	OH	H	OH	H
**32**	+	H	OH	OH	OH	H	OH	H	H	OH	H
**33**	−	h	OH	OMe	H	H	H	OH	H	OH	H
**34**	−	h	OH	OMe	H	H	H	OH	H	*O*-rut	H
**35**	−	h	OH	OMe	H	H	H	OH	H	*O*-neohesp	H
**36**	+	H	H	OH	H	H	4-Hp	H	H	OH	H
**37**	+	H	H	OH	H	H	4-Hp	OH	H	OH	H

+/− in the column π C2-C3 means presence or absence of double bond; Me—methyl; Bu—butenyl; Ac—acetyl; glc—glucose; gluc—glucuronide; ara—arabinoside; man—mannose; rut—rutinose; xyl—xylose; neohesp—neohesperidoside; 4-Hp—4-hydroxyphenyl.

**Table 2 cancers-16-00487-t002:** Cytotoxic activity of compounds **1**–**37** measured using MTT test. IC_50_ values are expressed as μM.

No.	A375			C32			Normal Cells		
	24 h	48 h	72 h	24 h	48 h	72 h	24 h	48 h	72 h
**1**	>200	31.9 ± 3.0	20.7 ± 1.5	>200	63.1 ± 4.7	31.8 ± 2.5	>200	>200	100–200
**2**	>200	>200	>200	>200	>200	>200	>200	>200	>200
**3**	>200	>200	>200	>200	>200	>200	>200	>200	>200
**4**	>200	>200	>200	>200	>200	>200	>200	>200	>200
**5**	>200	>200	>200	>200	>200	>200	>200	>200	>200
**6**	>200	38.2 ± 4.1	24.6 ± 1.6	>200	42.1 ± 3.5	13.1 ± 1.3	>200	69.7 ± 5.2	20.1 ± 1.4
**7**	>200	>200	37.1 ± 3.0	>200	60.5 ± 4.6	40.7 ± 2.9	>200	>200	>200
**8**	>200	>200	>200	>200	>200	>200	>200	>200	>200
**9**	>200	>200	>200	>200	>200	>200	>200	>200	>200
**10**	>200	>200	>200	>200	>200	>200	>200	>200	>200
**11**	>200	>200	>200	>200	>200	>200	>200	>200	>200
**12**	>200	>200	>200	>200	>200	>200	>200	>200	>200
**13**	>200	>200	>200	>200	>200	>200	>200	>200	>200
**14**	100–200	66.4 ± 4.5	34.4 ± 2.8	100–200	96.4 ± 7.0	58.9 ± 4.5	>200	>200	71.2 ± 4.5
**15**	86.0 ± 6.3	36.4 ± 2.0	26.4 ± 1.2	100–200	47.5 ± 3.5	23.5 ± 1.9	>200	>200	>200
**16**	100–200	33.16 ± 3.2	20.5 ± 2.2	100–200	39.2± 2.6	26.2 ± 2.8	>200	100–200	93.3 ± 8.0
**17**	97.9 ± 7.3	31.1 ± 2.9	20.5 ± 1.9	100–200	26.5 ± 1.3	13.9 ± 1.2	>200	78.0 ± 5.2	53.9 ± 3.6
**18**	100–200	100–200	75.8 ± 5.8	100–200	100–200	58.4 ± 3.5	100–200	100–200	55.8 ± 4.5
**19**	100–200	100–200	61.8 ± 3.1	100–200	100–200	100–200	>200	>200	100–200
**20**	94.2 ± 6.9	46.9 ± 3.5	21.5 ± 2.2	>200	>200	>200	>200	>200	>200
**21**	>200	>200	>200	>200	>200	>200	>200	>200	>200
**22**	>200	>200	>200	>200	>200	>200	>200	>200	>200
**23**	>200	>200	>200	>200	>200	>200	>200	>200	>200
**24**	>200	>200	74.4 ± 5.0	>200	>200	68.1 ± 4.6	>200	>200	36.1 ± 2.0
**25**	100–200	42.1 ± 3.0	25.1 ± 2.3	>200	100–200	37.7 ± 2.5	100–200	45.3 ± 2.3	35.7 ± 2.9
**26**	86.30 ± 5.9	68.5 ± 3.9	50.8 ± 4.0	>200	100–200	49.3 ± 3.6	100–200	68.5 ± 5.4	46.2 ± 3.3
**27**	>200	>200	>200	>200	>200	>200	>200	>200	>200
**28**	>200	57.0 ± 3.1	57.8 ± 4.8	>200	>200	>200	>200	>200	>200
**29**	>200	>200	>200	>200	>200	>200	>200	>200	>200
**30**	>200	>200	75.3 ± 4.8	>200	>200	>200	>200	>200	>200
**31**	>200	100–200	100–200	>200	100–200	100–200	100–200	80.3 ± 6.6	56.4 ± 4.7
**32**	>200	>200	>100–200	>200	>200	>200	>200	>200	>200
**33**	100–200	100–200	74.5 ± 6.4	>200	100–200	76.5 ± 6.3	>200	>200	>200
**34**	>200	>200	>200	>200	>200	>200	>200	>200	>200
**35**	>200	>200	>200	>200	>200	>200	>200	>200	>200
**36**	>200	>200	>200	>200	>200	>200	>200	>200	>200
**37**	93.1 ± 5.8	60.9 ± 4.2	24.2 ± 2.6	>200	>200	51.1 ± 3.5	>200	100–200	56.8 ± 4.1
**cPt ^A^**	15.6 ± 2.3	8.1 ± 1.1	8.8 ± 1.1	18.3 ± 2.3	6.6 ± 1.1	6.2 ± 1.1	100–200	40.4 ± 4.5	24.5 ± 1.5

Values are expressed as µM and represented as the mean ± standard deviation (SD) from at least three independent measurements; ^A^: cPt–cisplatin used as positive control.

**Table 3 cancers-16-00487-t003:** Cytotoxic activity of compounds **1**–**37** measured using NRU assay. IC_50_ values are expressed as μM.

No.	A375			C32			Normal Cells		
	24 h	48 h	72 h	24 h	48 h	72 h	24 h	48 h	72 h
**1**	64.5 ± 4.3	24.7 ± 2.4	21.2 ± 1.4	100–200	25.7 ± 1.6	22.4 ± 1.5	>200	87.8 ± 5.8	58.0 ± 3.2
**2**	>200	>200	>200	>200	>200	>200	>200	>200	>200
**3**	>200	>200	>200	>200	>200	>200	>200	>200	>200
**4**	>200	>200	>200	>200	>200	>200	>200	>200	>200
**5**	>200	>200	>200	>200	>200	>200	>200	>200	>200
**6**	58.1 ± 4.2	25.7 ± 1.5	23.9 ± 0.23	41.3 ± 3.8	13.1 ± 0.11	12.6 ± 1.2	>200	45.2 ± 2.6	21.4 ± 1.7
**7**	>200	56.7 ± 3.8	41.5 ± 3.3	>200	29.6 ± 2.0	23.0 ± 2.1	>200	100–200	100–200
**8**	>200	>200	>200	>200	>200	>200	>200	>200	>200
**9**	>200	>200	>200	>200	>200	>200	>200	>200	>200
**10**	>200	>200	>200	>200	>200	>200	>200	>200	>200
**11**	>200	>200	>200	>200	>200	>200	>200	>200	>200
**12**	>200	>200	>200	>200	>200	>200	>200	>200	>200
**13**	>200	>200	>200	>200	>200	>200	>200	>200	>200
**14**	52.4 ± 3.5	34.5 ± 2.3	33.1 ± 2.8	89.2 ± 7.8	44.7 ± 3.9	25.6 ± 2.4	100–200	65.7 ± 4.2	76.3 ± 4.7
**15**	53.3 ± 4.6	16.0 ± 1.1	13.6 ± 1.1	72.6 ± 5.5	18.2 ± 2.1	15.4 ± 1.1	>200	100–200	100–200
**16**	100–200	36.0 ± 3.5	27.4 ± 2.4	>200	19.9 ± 1.2	16.5 ± 2.1	>200	100–200	72.9 ± 5.6
**17**	42.6 ± 3.2	14.7 ± 1.2	14.6 ± 1.2	>200	14.7 ± 2.2	13.9 ± 2.1	>200	59.4 ± 5.2	43.5 ± 2.9
**18**	100–200	57.4 ± 4.6	47.7 ± 3.9	>200	62.8 ± 3.5	34.4 ± 2.9	>200	69.8 ± 5.8	51.1 ± 3.2
**19**	100–200	100–200	100–200	100–200	88.8 ± 7.0	71.9 ± 7.0	>200	>200	64.8 ± 5.0
**20**	>200	>200	>200	>200	>200	>200	>200	>200	>200
**21**	>200	>200	>200	>200	>200	>200	>200	>200	>200
**22**	>200	>200	>200	>200	>200	>200	>200	>200	>200
**23**	>200	>200	>200	>200	>200	>200	>200	>200	>200
**24**	100–200	84.7 ± 6.5	79.2 ± 7.3	>200	58.0 ± 3.5	47.5 ± 4.3	>200	92.6 ± 4.2	36.4 ± 0.23
**25**	>200	63.4 ± 4.8	43.7 ± 3.5	>200	55.4 ± 4.3	28.4 ± 2.5	>200	66.3 ± 5.6	38.4 ± 3.4
**26**	>200	100–200	78.1 ± 6.5	>200	100–200	65.1 ± 4.0	>200	78.7 ± 5.5	69.2 ± 4.6
**27**	>200	>200	>200	>200	>200	100–200	>200	>200	>200
**28**	>200	>200	>200	>200	>200	>200	>200	>200	>200
**29**	>200	>200	>200	>200	>200	>200	>200	>200	>200
**30**	>200	>200	>200	>200	>200	>200	>200	>200	>200
**31**	>200	>200	>200	>200	>200	>200	>200	>200	>200
**32**	>200	>200	>200	>200	>200	>200	>200	>200	>200
**33**	100–200	100–200	100–200	69.0 ± 5.1	58.3 ± 5.6	53.7 ± 4.0	>200	>200	100–200
**34**	>200	>200	>200	>200	>200	>200	>200	>200	>200
**35**	>200	>200	>200	>200	>200	>200	>200	>200	>200
**36**	>200	>200	>200	>200	>200	>200	>200	>200	>200
**37**	49.2 ± 3.2	32.3 ± 3.5	19.3 ± 2.2	100–200	31.5 ± 2.8	14.4 ± 2.1	>200	57.8 ± 4.0	38.6 ± 3.7
**cPt ^A^**	16.6 ± 2.1	14.4 ± 1.1	9.9 ± 1.1	15.7 ± 1.2	15.0 ± 1.2	13.8 ± 1.5	100–200	47.5 ± 4.9	24.4 ± 3.5

Values are expressed as µM and represented as the mean ± standard deviation (SD) from at least three independent measurements; ^A^: cPt—cisplatin as a positive control.

**Table 4 cancers-16-00487-t004:** The effect of compounds **1**, **6**, **15**–**17**, and **37** on DNA biosynthesis. IC_50_ values are expressed as μM.

	A375			C32			Normal Cells		
	24 h	48 h	72 h	24 h	48 h	72 h	24 h	48 h	72 h
**1**	87.7 ± 4.2	28.6 ± 1.8	23.4 ± 1.6	>200	27.0 ± 1.4	22.5 ± 1.2	187.5 ± 8.1	188.6 ± 9.5	>200
**6**	>200	38.5 ± 2.9	30.9 ± 1.8	>200	30.9 ± 2.2	24.5 ± 1.81	>200	>200	>200
**15**	>200	32.4 ± 2.4	31.4 ± 1.7	>200	20.0 ± 1.9	20.3 ± 1.2	174.9 ± 7.6	>200	>200
**16**	62.1 ± 3.9	29.7 ± 1.6	20.5 ± 2.2	38.1 ± 1.5	18.4 ± 1.4	12.8 ± 0.9	81.6 ± 5.8	64.6 ± 2.8	37.0 ± 2.5
**17**	>200	6.0 ± 0.2	5.34 ± 0.4	58.7 ± 3.8	9.1 ± 1.2	5.8 ± 3.1	>200	>200	87.9 ± 4.0
**37**	153.5 ± 10.8	28.0 ± 0.8	24.2 ± 1.5	25.4 ± 1.3	18.5 ± 0.7	19.1 ± 0.3	>200	>200	>200
**cPt**	3.8 ± 0.2	3.9 ± 0.5	3.4 ± 0.3	6.8 ± 0.4	5.7 ± 0.3	4.4 ± 0.4	7.1 ± 0.9	6.1 ± 0.5	6.1 ± 0.8

Values are expressed as µM and represented as the mean ± standard deviation (SD) from at least three independent measurements; cPt—cisplatin as positive control.

## Data Availability

The data presented in this study are available in this article and Appendix A.

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
