# Peer review of "Using Flavonoid Substitution Status to Predict Anticancer Effects in Human Melanoma Cancers: An In Vitro Study"

_cancers, 2024, doi:10.3390/cancers16030487_

Round 1

Reviewer 1 Report

Comments and Suggestions for Authors

Reviewer Comments for the manuscript: Flavonoid Substitution Status to Predict Anticancer Effects in 2 Human Melanoma Cancers: In Vitro Study

The authors in the current manuscript attempted to evaluate the anti-melanoma ability of a broad library of flavonoid derivatives. The authors also attempted to document the relationship between the existence of specific substituents on specific positions in the flavonoid core and the activation of either intrinsic and/or extrinsic apoptosis in addition to their ability to prevent DNA biosynthesis. In order to address the deliverables, the authors performed various assays including viability assay, Neutral Red assay, quantification of early and late stages of apoptosis induction, western immunoblots and immunostaining.

Introduction

Comment 1: Introduction is too short. The authors have to expand the introduction stating what are the current therapeutic options for the clinical management of malignant melanoma (ie what drugs are available, radiotherapy etc..) and why you chose flavonoids.

Comment 2: In the introduction section (line 48-49) the authors state that flavonoids are known for anticancer activity without explaining how and since the whole study emphasises on flavonoids, such information are essential.

Comment 3: The authors used two melanoma cell lines A375 and C32 and one fibroblast cell line. The authors should provide just a very brief description in the introduction (ie primary human malignant melanoma (A375) cells).

Comment 4: The authors have screened the anticancer activity of broad spectrum of polyphenolic compounds lines (56-70), which I would like to suggest their further subcategorization into for example flavanols, flavones, isoflavones, chalcones, flavononols etc, as this will enhance the conclusion output and the presentation of the compounds.

Comment 5: I would expect to see some key information regarding the (bio)chemistry of flavonoids ie. direct metal chelation (mainly iron which is incorporated into Fenton Chemistry) and ROS scavenging.

Materials and Methods

Line 86: The authors should briefly explain how the structure of the isolated flavonoids have been confirmed (ie mass spectrometry, NMR, XRAY crystallography)

Line 94: DMEM stands for Dulbecco’s Modified Eagle’s Medium and not for Dulbecco’s minimal essential medium

Line 96: what was the concentration of antibiotics added in the medium?

Comment 1: MTT assay (line 116). (State full name in the title of this section not the acronym form) The authors should provide details regarding the concentration range that have been used to screen the toxicity of the various flavonoids. Also, information regarding the cell population that has been used for the viability studies should be provided for both cells-lines. I would like to ask if the authors can state at this point that fibroblast have been used as control cell line in order to assessed the selectivity of the compounds. Line 122: concentration is needed for Sorensen’s glycine buffer.

Comment 2: NRU assay (line 129) (State full name in the title of this section not the acronym form)
line 132-133 the authors state the cells were treated with different concentrations of compounds. What compounds (provide either compound number or compound name)? also at what concentrations? I was also wonder if all three cell lines were grown at the same density (could you please clarify this? Since A375 and fibroblasts have considerably different size and growth capacity).

Comment 3: Proliferation Assay (line 139)
line 141: normal cells? Did the authors meant fibroblast? Is so please substitute normal cells with fibroblasts.

Comment 4: Apoptosis Assay (line 150)
line 153: selected flavonoids (which are these? Please provide either number or full name).

Comment 5: Prior to western immunoblotting I would like the authors to include another section describing the process of protein extraction lysis and quantification since that information are missing. In the current state, there is no information regarding the sonication duration, lysis buffer used nor the amount of protein loaded. Also, there is not any information regarding the standardisation of protein concentration ie. did the authors used BCA assay, Bradford reagent and if so at what range of concentrations?

Comment 6: Immunofluorescence Microscopy (line 171)
line 173-174: The authors should provide details regarding what flavonoids have they chosen (either number or full name) and at what concentrations?

Results

With respect to the various cytotoxicity assays I would like the authors to include the different cytotoxicity curves (for all compounds and cell lines) in a Supplementary material.

Despite the fact that authors claimed into materials and methods section that they performed statistical analysis there is not any statistical information in the manuscript.  

Figure 2: I believe that is better to remove the percentages from flow cytometry results and allow the histograms with a single explanation of the four populations (which is not explained). Present the results on a separate graph for a better clarity.

Figure 3: The authors should include a graph either in the actual manuscript or in a supplementary material showing the western blot bands intensity quantification.

 Figure 4 and 5: The authors should include a graph either in the actual manuscript or in a supplementary material showing the quantitated results from the immunostaining results.  

Figure 4 and 5 can be merged in one figure.

Flow cytometry/ Western immunoblotting and immunostaining data for fibroblasts should also be presented in the manuscript as a way of validating the selectivity. Cell viability data should not be the only criteria for demonstrating the selectivity of flavonoids for melanoma cells.

Discussion

Line 150-154 Why is that observation? Can the authors prove/correlate this with any SARs data from in silico studies (ie simulations) with particular protein?
Figure 7: It is still unclear how these structural modifications can be related to either the activation of intrinsic and/or extrinsic apoptosis. More evidences should be provided on this based NOT on the activity/ expression of apoptotic related protein but simulation data in order to link the interactions that can be developed between the modified sites and the protein(s) of interest. As I stated above, activation of apoptosis at various extends can indeed be triggered through the alteration of substituents, mainly by regulating ROS levels (either by inducing ROS or by scavenging ROS) (ROS are directly linked with extrinsic apoptosis). On the other hand, the OCH3 functionality can inhibit DNA biosynthesis. How this modification can affect this observation? Does it increase the intercalating ability of polyphenols?

Also, there wasn’t any explanation on why there is selectivity of the selected compounds in melanoma cell lines and not on fibroblasts (please provide).

I would encourage the authors to use the selected compounds (not the whole library) and quantify the ROS profile in all 3 cell lines (before and after the addition). Then correlate the various substituents with the ROS levels and apoptotic induction. (This approach may answer my question above). In such study, the quantification or ROS levels in general is of high importance particularly when you use polyphenolic compounds.  

Author Response

Point-by-point response to the Reviewer #1 Comments and Suggestions for Authors:

The authors in the current manuscript attempted to evaluate the anti-melanoma ability of a broad library of flavonoid derivatives. The authors also attempted to document the relationship between the existence of specific substituents on specific positions in the flavonoid core and the activation of either intrinsic and/or extrinsic apoptosis in addition to their ability to prevent DNA biosynthesis. In order to address the deliverables, the authors performed various assays including viability assay, Neutral Red assay, quantification of early and late stages of apoptosis induction, western immunoblots and immunostaining.

Introduction

Comment 1: Introduction is too short. The authors have to expand the introduction stating what are the current therapeutic options for the clinical management of malignant melanoma (ie what drugs are available, radiotherapy etc..) and why you chose flavonoids.

Response: As The Reviewer suggested more information’s have been added in the 42-51 lines.

Comment 2: In the introduction section (line 48-49) the authors state that flavonoids are known for anticancer activity without explaining how and since the whole study emphasises on flavonoids, such information are essential.

Response: As The Reviewer suggested more information’s have been added in the 53-61 lines.

Comment 3: The authors used two melanoma cell lines A375 and C32 and one fibroblast cell line. The authors should provide just a very brief description in the introduction (ie primary human malignant melanoma (A375) cells).

Response: As The Reviewer suggested more information’s have been added in the 66 and 67 lines.

Comment 4: The authors have screened the anticancer activity of broad spectrum of polyphenolic compounds lines (56-70), which I would like to suggest their further subcategorization into for example flavanols, flavones, isoflavones, chalcones, flavonols etc, as this will enhance the conclusion output and the presentation of the compounds.

Response: As The Reviewer suggested more information’s have been added in the 86-92 lines.

Comment 5: I would expect to see some key information regarding the (bio)chemistry of flavonoids ie. direct metal chelation (mainly iron which is incorporated into Fenton Chemistry) and ROS scavenging.

Response: As The Reviewer suggested more information’s have been added in the 56-64 lines.

Materials and Methods

Line 86: The authors should briefly explain how the structure of the isolated flavonoids have been confirmed (ie mass spectrometry, NMR, XRAY crystallography)

Response: The structures of isolated compounds are described in citated manuscripts: 13-22. the remaining compounds were purchased.

Line 94: DMEM stands for Dulbecco’s Modified Eagle’s Medium and not for Dulbecco’s minimal essential medium

Response: The information’s have been corrected in the 116 line.

Line 96: what was the concentration of antibiotics added in the medium?

Response: The information’s have been added in the 117 and 118 lines.

Comment 1: MTT assay (line 116). (State full name in the title of this section not the acronym form) The authors should provide details regarding the concentration range that have been used to screen the toxicity of the various flavonoids. Also, information regarding the cell population that has been used for the viability studies should be provided for both cells-lines. I would like to ask if the authors can state at this point that fibroblast have been used as control cell line in order to assessed the selectivity of the compounds. Line 122: concentration is needed for Sorensen’s glycine buffer.

Response: MTT assay section was corrected according to Reviewer suggestions. The information’s have been added in the 137, 140-146 lines.

Comment 2: NRU assay (line 129) (State full name in the title of this section not the acronym form) line 132-133 the authors state the cells were treated with different concentrations of compounds. What compounds (provide either compound number or compound name)? also at what concentrations? I was also wonder if all three cell lines were grown at the same density (could you please clarify this? Since A375 and fibroblasts have considerably different size and growth capacity).

Response: NRU assay section was corrected according to Reviewer suggestions (152-157 lines). All cell lines were seeded at 1 × 104 cells per well. We agree with the Reviewer that both melanoma cells have significantly smaller size than fibroblasts. However, their population doubling time is much faster comparing with fibroblasts, and 18 hours after plating almost all melanoma cells divided, while the first fibroblast division was delayed. Furthermore, even after this time, only a part of population of fibroblasts divides during the 24-hour period. In contrast, each subsequent cell division of melanoma covers nearly entire population and requires not more than 24 hours. As a result, cell confluence just before the addition of the tested compounds were 10-15% for melanoma and 30% for fibroblasts and it gained after 72 hours almost full confluence in vehicle-treated populations of melanoma cells (with no difference between A375 and C32) and about 80% in vehicle-treated population of fibroblasts.

Comment 3: Proliferation Assay (line 139)

line 141: normal cells? Did the authors meant fibroblast? Is so please substitute normal cells with fibroblasts.

Response: The information’s have been corrected in the 165 line.

Comment 4: Apoptosis Assay (line 150)

line 153: selected flavonoids (which are these? Please provide either number or full name).

Response: The information’s have been added in the 177-178 lines.

Comment 5: Prior to western immunoblotting I would like the authors to include another section describing the process of protein extraction lysis and quantification since that information are missing. In the current state, there is no information regarding the sonication duration, lysis buffer used nor the amount of protein loaded. Also, there is not any information regarding the standardisation of protein concentration ie. did the authors used BCA assay, Bradford reagent and if so at what range of concentrations?

Response: We decided to insert this information’s into 2.5 Western Immunoblot section. Range of protein concentration used in the protein measurement method was 10-100 μg/mL.

Comment 6: Immunofluorescence Microscopy (line 171)

line 173-174: The authors should provide details regarding what flavonoids have they chosen (either number or full name) and at what concentrations?

Response: The information’s have been added in the 203-205 and 213-214 lines.

Results

With respect to the various cytotoxicity assays I would like the authors to include the different cytotoxicity curves (for all compounds and cell lines) in a Supplementary material.

Response: Cytotoxicity curves were attached in Supplementary Figures S1 and S2.

Despite the fact that authors claimed into materials and methods section that they performed statistical analysis there is not any statistical information in the manuscript. 

Response: Statistical analysis was attached in Figure 2 and supplementary Figures 1-4 in the corrected manuscript.

Figure 2: I believe that is better to remove the percentages from flow cytometry results and allow the histograms with a single explanation of the four populations (which is not explained). Present the results on a separate graph for a better clarity.

Response: As The Reviewer suggested Figure 2 has been corrected.

Figure 3: The authors should include a graph either in the actual manuscript or in a supplementary material showing the western blot bands intensity quantification.

Response: The graph was added as the Figure S3.

 Figure 4 and 5: The authors should include a graph either in the actual manuscript or in a supplementary material showing the quantitated results from the immunostaining results. 

Response: The graph was added as the Figure S4.

Figure 4 and 5 can be merged in one figure.

Response:  As The Reviewer suggested Figure 4 and 5 have been merged.

Flow cytometry/ Western immunoblotting and immunostaining data for fibroblasts should also be presented in the manuscript as a way of validating the selectivity. Cell viability data should not be the only criteria for demonstrating the selectivity of flavonoids for melanoma cells.

Response: We appreciate this suggestion, but in this manuscript, we choose the viability assays to assess the selectivity of compounds and, in order to make the greatest effort to obtain reliable results, we used two methods (MTT and NRU assays).

Discussion

Line 150-154 Why is that observation? Can the authors prove/correlate this with any SARs data from in silico studies (ie simulations) with particular protein?

Response: The information’s have been added in the section 4, 153-169 lines.

Figure 7: It is still unclear how these structural modifications can be related to either the activation of intrinsic and/or extrinsic apoptosis. More evidences should be provided on this based NOT on the activity/ expression of apoptotic related protein but simulation data in order to link the interactions that can be developed between the modified sites and the protein(s) of interest. As I stated above, activation of apoptosis at various extends can indeed be triggered through the alteration of substituents, mainly by regulating ROS levels (either by inducing ROS or by scavenging ROS) (ROS are directly linked with extrinsic apoptosis). On the other hand, the OCH3 functionality can inhibit DNA biosynthesis. How this modification can affect this observation? Does it increase the intercalating ability of polyphenols?

I would encourage the authors to use the selected compounds (not the whole library) and quantify the ROS profile in all 3 cell lines (before and after the addition). Then correlate the various substituents with the ROS levels and apoptotic induction. (This approach may answer my question above). In such study, the quantification or ROS levels in general is of high importance particularly when you use polyphenolic compounds. 

Response: Subsequent stages of research are necessary to assess the exact mechanisms of action of individual compounds. Due to methodological limitations, it is impossible to perform ROS determination using cells (not enough amount of compounds).

Reviewer 2 Report

Comments and Suggestions for Authors

This manuscript presents the various flavonoids (37 flavonoids) containing anticancer activity to understand the structure-activity relationships. The writing is straightforward and the experimental/theoretical approaches of the presented research make it attracted to the readership of the Cancers; however, major revision of the manuscript is necessary prior to publication.

Comments:

1.        For the cell studies, the authors should provide more detailed information of cell lines.

2.        In Table 2 and 3, the authors summarize the IC50 values of the flavonoids and cisplatin (as positive control) obtained upon treatment to A375, C32, and normal cells after 24, 48, and 72 h.

                       i.              Based on the results, none of the flavonoids show better anticancer effect than cisplatin. Then, how can we improve the structure of flavonoids to have effective anticancer ability?

                    ii.              Compound 5 and 6 look toxic to normal cells as well.

                  iii.              With MTT assay, compound 32 shows its IC50 values to A375 cells as 51.1, 35, and >200 uM. Are these results reproducible?

                   iv.              For the proliferation assay, why did the authors selected compound 1, 6, 15, 16, 17, and 37 for testing?

                     v.              How the longer incubation of some compounds showed less toxicity? For example, compound 30 and 32 treated to A375 cells.

3.        In Figure 3, the authors did not present the “caspase-8” bands from C32. Were those bands never seen? Is there any specific reason?

4.        For immunostaining studies, how much compounds were treated to the cells?

                       i.              Did the death of cells affect the results?

                    ii.              What fluorophores were applied to present caspases, nuclei, and F-actin? The information should be described in Materials and Methods section.

                  iii.              What were the excitation/emission wavelength? Were there any interference by flavonoids?

Comments on the Quality of English Language

The grammatical errors and typos should be fixed.

Author Response

Point-by-point response to the Reviewer #2 Comments and Suggestions for Authors:

This manuscript presents the various flavonoids (37 flavonoids) containing anticancer activity to understand the structure-activity relationships. The writing is straightforward and the experimental/theoretical approaches of the presented research make it attracted to the readership of the Cancers; however, major revision of the manuscript is necessary prior to publication.

Comments:

  1. For the cell studies, the authors should provide more detailed information of cell lines.

Response: A the Reviewer suggested more detailed information are added in the Introduction section.

  1. In Table 2 and 3, the authors summarize the IC50 values of the flavonoids and cisplatin (as positive control) obtained upon treatment to A375, C32, and normal cells after 24, 48, and 72 h.
  2. Based on the results, none of the flavonoids show better anticancer effect than cisplatin. Then, how can we improve the structure of flavonoids to have effective anticancer ability?

Response: The flavonoids used are less toxic to normal cells than cisplatin. The in vitro studies highlight the importance of the flavonoid core with hydroxyl groups.

  1. Compound 5 and 6 look toxic to normal cells as well.

Response: We wish to apologize the Reviewer for the mistakes in table 2. The table data were carefully checked and all mistakes were corrected.

iii. With MTT assay, compound 32 shows its IC50 values to A375 cells as 51.1, 35, and >200 uM. Are these results reproducible?

Response: We wish to apologize the Reviewer for the mistakes in table 2. The table data were carefully checked and all mistakes were corrected. Among flavonoid glycosides tested only cynaroside exerted significant cytotoxicity in MTT assay and NRU assay. The determinations were performed in at least 3 replicates.

  1. For the proliferation assay, why did the authors selected compound 1, 6, 15, 16, 17, and 37 for testing?

Response: We selected compounds 1, 6, 15, 16, 17, and 37 based on the cytotoxic tests and their the most potent activity.

  1. How the longer incubation of some compounds showed less toxicity? For example, compound 30 and 32 treated to A375 cells.

Response: We wish to apologize the Reviewer for the mistakes in table 2. The table data were carefully checked and all mistakes were corrected. Among flavonoid glycosides tested only cynaroside exerted significant cytotoxicity in MTT assay and NRU assay.

In Figure 3, the authors did not present the “caspase-8” bands from C32. Were those bands never seen? Is there any specific reason?

Response: There are no literature showing expression of procaspase-8 in the C32 line determined by western immunoblot. Probably procaspase-8 is not expressed or for other reasons procaspase-8 is not detected by the antibody in these cells. However, this antibody works well in a dozen of human cell lines and only in C32 cells we have never obtained bands.

  1. For immunostaining studies, how much compounds were treated to the cells?
  2. Did the death of cells affect the results?

Response: We have made great efforts to ensure that during washing there is as little loss of loosely attached cells that are late apoptotic. These cells show significant expression of active executioner caspases.

  1. What fluorophores were applied to present caspases, nuclei, and F-actin? The information should be described in Materials and Methods section.

Response: Fluorophores used in fluorescence microscopy were Alexa Fluor 594, Hoechst 33342 and Atto 488 for visualization of active caspases, nuclei and F-actin, respectively. This information is available in 2.6. Immunofluorescence microscopy section.

iii. What were the excitation/emission wavelength? Were there any interference by flavonoids?

Response: The spectra are as follows: Alexa Fluor 594 (excitation 555 nm, emission 645 nm), Hoechst 33342 (excitation 360 nm, emission 435 nm), Atto 488 (excitation 488 nm, emission 515 nm). Flavonoids did not interfere with the fluorochromes used.

Round 2

Reviewer 1 Report

Comments and Suggestions for Authors

The authors addressed all my concerns/recommendations

Very well done!

Reviewer 2 Report

Comments and Suggestions for Authors

All the my concerns were cleared. The revised manuscript is suitable for publication.